# Ketosis Alters Transcriptional Adaptations of Subcutaneous White Adipose Tissue in Holstein Cows during the Transition Period

**DOI:** 10.3390/ani12172238

**Published:** 2022-08-30

**Authors:** Mao Ning, Yihan Zhao, Zhixin Li, Jie Cao

**Affiliations:** 1College of Veterinary Medicine, China Agricultural University, Beijing 100193, China; 2Ningxia Center for Animal Disease Prevention and Control, Yinchuan 750001, China

**Keywords:** perinatal period, ketosis, sWAT, glyceroneogenesis, inflammatory response

## Abstract

**Simple Summary:**

Ketosis in dairy cows is one of the common nutritional metabolic diseases during the perinatal period. We conducted the present subcutaneous white adipose tissue transcriptome-based study to investigate the pathogenesis of ketosis in dairy cows. The transcriptome results showed that the expression of PCK1 and PCK2, key enzymes of the glyceroneogenesis, was simultaneously downregulated post-partum compared to pre-partum, possibly resulting in impaired lipid storage in the adipose tissue of ketotic cows. Integrating the GO enrichment analysis and published gene expression traits, the inflammatory response biological process inhibiting PCK1 and PCK2 expression in adipose tissue may be important in the development of ketosis in dairy cows. The findings of the present study provided important clues for the therapeutic and preventive targets for ketosis.

**Abstract:**

Ketosis is a common nutritional, metabolic disease during the perinatal period in dairy cows characterized by elevated blood β-hydroxybutyrate (BHBA). In this study, RNA sequencing (RNA-seq) was performed to investigate adaptive changes in adipose tissue during the perinatal period of dairy cows. Blood and tailhead subcutaneous white adipose tissue (sWAT) were obtained from ketotic cows (Ket = 8, BHBA ≥ 1.4 mmol/L) and non-ketotic cows (Nket = 6, BHBA < 1.4 mmol/L) 21 d pre-partum and 10 d post-partum. Compared with pre-partum, decreased lipid synthesis due to down-regulation of PCK1 may be in a strong association with clinical ketosis. Simultaneously, PCK2 was downregulated in the Ket postnatally compared to its expression prenatally, and the expression of PCK2 was 2.7~4.2 times higher than that of PCK1, implying a more severe lipid storage impairment in the Ket. Moreover, compared to pre-partum, the upregulated differentially expressed genes post-partum in the Ket were enriched in the inflammatory response biological process. The higher expression of TNC (tenascin C) in the post-partum Ket relative to the Nket suggested that the adipose tissue of ketotic cows might also be accompanied by tissue fibrosis. Notably, pre-partum CD209 was higher in the Ket than in the Nket, which might be used as a candidate marker for the pre-partum prediction of ketosis. Combined with published gene expression traits, these results suggested that inflammation leads to a more widespread downregulation of the lipid synthesis gene network in adipose tissue in ketotic cows. Additionally, sWAT in post-partum cows with ketosis might also be accompanied by tissue fibrosis which could make the treatment of ketosis more difficult.

## 1. Introduction

Ketosis (blood β-hydroxybutyrate (BHBA) ≥ 1.4 mmol/L) is a common nutritional metabolic disease in dairy cows during the perinatal period (from 21 d pre-partum to 21 d post-partum) [1,2]. This disease is associated with negative energy balance (NEB), lipidosis, hormonal changes, and increased concentration of blood hydroxybutyrate while blood glucose concentration is decreased [3]. Ketosis can affect the reproductive performance of dairy cows, reduce their lactation performance, and may be accompanied by other diseases, thus bringing huge economic losses to farms. During the third trimester of pregnancy and early lactation, nutritional requirements for fetal growth and milk synthesis increase dramatically. However, cows cannot meet quickly meet these energy requirements through feeding in the post-partum period. Therefore, most cows enter negative energy balance during the perinatal period and often take several weeks to recover. To compensate for low energy in the perinatal period, cows mobilize body fat for energy, as shown by increased concentration of plasma non-esterified fatty acids (NEFA) and concentration of blood BHBA that are proportional to the negative energy balance (NEB) [4,5,6,7]. When NEFA is fully oxidized in the liver, it is converted into carbon dioxide and water; when NEFA is incompletely oxidized in the liver, it is converted into ketone bodies [8,9,10]. Ketone bodies can supply energy to extrahepatic tissues (e.g., muscle and brain) and are an adaptation in the perinatal period of the cow. However, once the blood BHBA exceed the threshold (BHBA ≥ 1.4 mmol/L), they can cause ketosis and suppress immunity and feed intake, thus threatening the health of cows.

Adipose tissue is a major part of the energy store in mammals. Adipose tissue undergoes tremendous adaptive changes during the perinatal period. The net output of NEFA in perinatal adipose tissue depends on the dynamic balance of lipolysis and lipogenesis. Previous studies have shown that perinatal adipose tissue change in dairy cows is a complex process, which includes not only the activation of lipid metabolism pathway in response to nerve and hormone stimulation, but also inflammatory response, immune cell migration, proliferation of some cellular components of stromal blood vessels, and changes of extracellular matrix [11]. Adipose tissue remodeling will further affect lipid metabolism in adipose tissue by changing the secretion pattern of adipose factors secreted by adipose tissue, and weaken the ability of AT to buffer excessively high concentration of circulating NEFA, thus increasing the risk of metabolic and inflammatory diseases [12]. However, the unique metabolic challenges that dairy cow adipose tissue faces during the transition limit the scope of extrapolating results from adipose tissue biology and human rodent model studies. At present, there is still a gap in our understanding of dairy cows’ perinatal adipose tissue. Therefore, the changes of dairy cows’ perinatal adipose tissue still need to be further studied.

Studies on the subcutaneous white adipose tissue (sWAT) in perinatal dairy cows have found that the mRNA expression of genes that control adipogenesis and key enzymes in the process of adipogenesis is sharply decreased in the early lactation stage. The activity of lipogenic enzymes is controlled by transcriptional mechanisms and affected by the availability of energy [13]. Negative energy balance at the beginning of lactation leads to a pronounced decrease in the expression and activity of genes encoding proteins of the de-novo lipogenic, and glycerol-3-phosphate pathways. Reduced lipogenic activity in AT may contribute to the increase in circulating FFA levels in the immediate postpartum period. At the same time, the mRNA expression of key lipolysis enzymes was also down-regulated after delivery while lipolysis is also primarily modulated by posttranscriptional control mechanisms and the rate of phosphorylation is increased post-partum [14]. The broadly decreased transcriptional regulation of the lipogenic gene network suggests that decreased lipogenesis is an important contributor to NEFA release from sWAT postpartum. In other words, the downregulation of the lipogenesis gene network, rather than an increase in lipolysis, may be crucial for increasing the NEFA levels in post-partum blood [15,16,17,18]. Therefore, this article will focus on the relationship between downregulation of adipogenesis and high NEFA levels post-partum. However, the studies could not explain why NEFA levels in the blood are higher in post-partum ketotic cows than in non-ketotic cows. In recent years, with the rise of transcriptomic technologies, it is possible to provide a more comprehensive picture of the study subject. Therefore, the transcriptome has become an important tool for exploring disease mechanisms. Mellouk (2019) identified the lipid metabolic gene network of adipose tissue as a key network affected by negative energy balance [19]. Because negative energy balance is a prerequisite for the development of ketosis, studies targeting the lipid metabolic network in ketosis adipose tissue are still necessary.

Here, we reported for the first time the adaptive changes at the transcriptome level in sWAT during the perinatal period in cows with ketosis. Although the effects of parturition and the initiation of lactation on the sWAT transcriptome play a dominant role, the pathogenesis of ketosis in cows can be better understood through this study, which can provide clues to finding therapeutic and preventive targets for ketosis.

## 2. Materials and Methods

### 2.1. Animal Model

The longitudinal cohort test was conducted on a commercially intensive farm (Anhui Province, China). Holstein cows (*n* = 30) were randomly selected based on the following conditions: 1. the cows were in the dry period; 2. the cows were multiparous (parity 2~5); 3. the cows were overconditioned with BCS ≥ 3.5 (their perinatal physical condition scores were evaluated by three experienced veterinarians with the mode taken as the final result); 4. blood BHBA of all cows was lower than 1.4 mmol/L on day 21 pre-partum (relative to the expected date of delivery). All cows were fed a fresh total mixed ration (TMR) thrice a day and were guaranteed to have 3–5% leftovers each time. They had free access to water throughout the day. All experimental operations were carried out in the morning before feeding. Blood was collected from coccygeal vein on day 21 pre-partum (d −21), day 5 (d +5) and day 10 (d +10) post-partum. The BHBA and glucose were immediately measured with a hand-held blood ketone meter (FreeStyle Optium Neo H-ketone Ltd.; Abbot Diabetes Care meter, Witney, Oxon, UK). The cows were divided into two groups according to their BHBA on d +5 and d +10; those with BHBA < 1.4 mmol/L on d +5 and d +10 were placed in the non-ketotic (Nket) cows, and the others were placed in the ketotic (Ket) cows. Most of the ketosis group belong to subclinical ketosis without obvious clinical symptoms.

### 2.2. Differential Gene Expression Analysis

After excluding cows with post-partum diseases, such as retained fetal membranes and abomasum displacement during tracking, the final results of transcriptome analysis were obtained from 14 cows (Ket = 8; Nket = 6).

#### 2.2.1. Fat Collection

Subcutaneous white adipose tissue (sWAT) samples were obtained from the tailhead on d –21 pre-partum and d +10 post-partum. At the tailhead area, a 10 cm × 10 cm surgical field was prepared for shaving and disinfection, 15 mL of 2% lidocaine was injected subcutaneously in the shaved area, and a surgical incision of about 2 cm was cut, then pick up the adipose tissue with tweezers, cut 1~2 g of adipose tissue with a scalpel, suture the wound with 2~3 needles, and finally apply the ointment to the wound and monitor the postoperative wound recovery. Sutures were removed after 14 d. Caudal sWAT samples collected from dairy cows were fixed with RNAlater^TM^ (Invitrogen, Thermo Fisher Scientific, Vilnius, Lithuania), and sent to Beijing Novogene Co., Ltd., (Beijing, China) for sWAT RNA extraction and sequencing. The databases have been submitted to the GenBank databases under accession number: PRJNA841800 and the URL is: https://dataview.ncbi.nlm.nih.gov/object/PRJNA841800?reviewer=iqedfkatp6siru4rhoosnehcli, (accessed on 7 July 2022).

#### 2.2.2. RNA Seq Analysis

The purity, concentration, and integrity of the extracted RNA were determined using a Nanospectrophotometer Drop 1000 (Thermo Scientific, Wilmington, DE, USA) and an Agilent Bioanalyzer 2100 system (Agilent Technologies, Santa Clara, CA, USA). The A260/A280 of mRNA in all samples was 1.9~2.1, and the RNA integrity number (RIN) was ≥8. Second-generation sequencing technology was used in this study. Initially, double-end sequencing was performed on an Illumina NovaSeq 6000 sequencing platform with a 150 bp read length. Then, FastQC v0.11 (www.bioinformatics.babraham.ac.uk/projects/fastqc/ accessed on 3 December 2020) was used for quality control, and the filtered clean data were mapped to the bosTau9 reference genome for alignment with STAR v2.7.5b, with an average mapping rate of 96.15%. Finally, the reads were quantified with RSEM v1.3.2 to construct a count matrix and a transcripts per million (TPM) matrix.

The count matrix was imported into R v4.0.5, and the genes with the top 5000 median absolute deviation (MAD) were selected for principal component analysis (PCA). The transcriptome was analyzed for differential expression with the software package DESeq2 v1.32.0, and genes with LFC (Log2FoldChange) > 1 or LFC < −1, Padj < 0.05 were defined as differentially expressed genes (DEG). After the genes were captured, they were imported into DAVID v6.8 (https://david.ncifcrf.gov/ accessed on 4 February 2021) for enrichment analysis; false discovery rate (FDR) ≤ 0.05 was used to determine significant enrichment.

### 2.3. Statistical Analysis

Biochemical data were analyzed for statistical differences with the SPSS R26.0.0.0 software (IBM Corp., Armonk, NY, USA). All biochemical data represents in the form of average ± standard error (LSM ± SEM), and the significant determination is * *p* < 0.05, ** *p* < 0.01.

## 3. Results

### 3.1. Serum Biochemical Parameters

Before parturition, blood BHBA was lower than 1.4 mmol/L in both groups, but blood BHBA was slightly higher in Nket than in Ket (*p* < 0.05); there was no significant difference in blood glucose (*p* > 0.05). Comparing prenatally, blood BHBA was elevated in both Ket and Nket on day 5 post-partum (*p* < 0.05), but blood BHBA was higher in Ket than in Nket on both day 5 and 10 post-partum, and both were higher than 1.4 mmol/L (*p* < 0.01). In contrast to prenatal, blood glucose tended to decrease on day 5 post-partum in both Ket and Nket, but only Ket had significantly lower glucose on day 5 post-partum than pre-partum (*p* < 0.05) (Figure 1).

### 3.2. Differential Expression Analysis

The results of the PCA showed that pre-partum samples were not separated, and the effect of parturition/initiation of lactation on adipose tissue transcriptome was dominant (Figure 2A). The number of captured DEGs corroborated the PCA results. Four DEGs (AMPD3, LHFPL4, CD209, and ST8SIA5) were upregulated pre-partum in the Ket relative to the expression levels in the NKet; 23 DEGs were upregulated, and 6 DEGs (GPR63, SLPI, EXTL1, SRCIN1, ADAMTS16, and GSTA1) were downregulated post-partum in the Ket relative to the gene expression in the Nket (Figure 2B). There were 532 upregulated DEGs, and 781 downregulated DEGs in the Ket in the post-partum period relative to the gene expression in the pre-partum period (Figure 2C); 355 DEGs were upregulated, and 496 DEGs were downregulated in the Nket in the post-partum period relative to the gene expression in the pre-partum period (Figure 2C). The results of GO enrichment analysis showed that the DEGs upregulated in the Ket post-partum were enriched in inflammatory response and cytokine production (FDR ≤ 0.05) (Figure 2D, Appendix A).

Phosphoenolpyruvate carboxykinase (PEPCK), a key enzyme of glyceroneogenesis, has two isoforms, which are encoded by PCK1 (phosphoenolpyruvate carboxykinase1) and PCK2 (phosphoenolpyruvate carboxykinase2). The sWAT PCK1 was downregulated in the post-partum period (Table 1). PCK2 in the Ket was also downregulated (*p* < 0.01) in the post-partum period, and the TPM of PCK2 was 2.7 and 4.3 times higher than that of PCK1 in the pre-partum and post-partum period, respectively. Genes in the lipolytic network were not significantly different (*p* > 0.05) at the two-time points and between the two groups (Table 1).

CD209 (cluster of differentiation 209) was upregulated in the Ket relative to its level in the Nket in the pre-partum period; TNC (tenascin C) was upregulated in the Ket relative to its level in the Nket in the post-partum period (Table 1).

## 4. Discussion

Holstein cows tracked in this study were divided into ketotic (Ket, *n* = 8) and non-ketotic (Nket, *n* = 6) cows based on BHBA on days 5 and 10 post-partum (threshold: 1.4 mmol/L). The sWAT was collected from these cows on day 21 pre-partum and day 10 post-partum for transcriptome differential expression analysis.

The concentration of NEFA, BHBA, and glucose in blood can indicate the negative energy balance of the body. Glucose concentration represents the energy state of the body, BHBA reflects the degree of incomplete lipid oxidation in the liver, and NEFA concentration can reflect the degree of body fat mobilization of the body [20,21]. Consistent with previous research results [22,23,24], BHBA of cows with ketosis was significantly higher than that of healthy cows within 10 days after delivery, and blood glucose level was lower than that of healthy cows, indicating that cows with ketosis were in a more serious negative energy balance or had insufficient ability to adapt to negative energy balance in the short term. Adipose tissue is a plastic organ. It is an important regulator of systemic metabolism in response to animal energy state [25]. Severe and persistent negative energy balance in ketosis cows results in changes in adipose tissue size and adipokine secretion patterns [26], thus the regulation of glucose and lipid metabolism in adipose tissue may be more disorganized than in healthy cows.

The two most basic metabolic activities of WAT, i.e., lipogenesis and lipolysis, regulate the net output of NEFA [27,28]. We found that post-partum serum NEFA increased significantly in both groups, which might also be related to the downregulation of WAT lipogenesis besides lipolysis, because PCK1, one of the key enzymes of glyceroneogenesis, was downregulated in the post-partum period. Our results were consistent with those of previous studies [15,17]. In this study, we did not find differential expression of the enzymes crucial to sWAT lipolysis (ATGL and HSL). However, lipolysis is also controlled by phosphorylation, so the status of lipolysis cannot be rigorously assessed [15,16,17]. Thus, the increase in post-partum plasma NEFA might be due to a decrease in lipogenesis as well.

Additionally, serum NEFA was higher in ketotic cows than in non-ketotic cows post-partum, which indicated that ketotic cows had more severe lipogenesis disorders. Glyceroneogenesis is a major pathway accounting for approximately 90% of triglyceride glycerol synthesis in adipocytes, and its key enzyme PEPCK has two isoforms, cytoplasmic PEPCK-C (encoded by the gene PCK1) and mitochondrial PEPCK-M (encoded by the gene PCK2) [29,30]. To the best of our knowledge, this was the first time that the expression of sWAT PCK2 was found to be downregulated in post-partum ketotic cows. The relative expression of PCK1 and PCK2 is species-specific [31]. Rat PCK2, which accounts for only 10% of the total PCK, plays an important role in the storage of lipids in WAT [32,33]. In this study, the expression of sWAT PCK2 in dairy cows was 2.7~4.2-fold higher than that of PCK1, and PCK1 and PCK2 were also simultaneously downregulated in the Ket in the post-partum period. Therefore, we speculated that PCK2 might be an important target of dysregulated adipose tissue mobilization in the post-partum period in dairy cows with ketosis.

The sWAT inflammatory response biological process in the Ket was upregulated postnatal compared to prenatal, relative to that in the NKet. Inflammation can inhibit the expression of PCK1 and PCK2 [29]. Thus, inflammation might contribute to post-partum dysregulation of lipid metabolism in ketotic cows. CD209 is a polarization marker for M2 macrophages, and its activation can reduce the inflammation of the adipose tissue [34,35]. In this study, we found that the expression level of CD209 was higher in the ketotic cows than in the non-ketotic cows prenatally, which indicated that sWAT in ketotic cows had become anti-inflammatory on day −21 pre-partum, and CD209 might be a candidate marker for predicting ketosis in the pre-partum adipose tissue.

Additionally, sWAT inflammation in the Ket might also be related to tissue fibrosis through TNC (tenascin C). TNC belongs to the damage-associated molecular pattern family, which is not only pro-inflammatory, but also involved in the fibrotic response of the extracellular matrix (ECM) [36,37,38,39]. Excessive accumulation of ECM disrupts the homeostasis of adipose tissue, causing adipose tissue inflammation and metabolic disturbances [36]. Thus, we hypothesized that sWAT fibrosis in the ketotic cows mediated by TNC might be responsible for the low rate of cure and poor prognosis of ketosis.

## 5. Conclusions

In this study, the transcriptome analysis of sWAT not only showed the adaptive changes of adipose tissue in dairy cows during the perinatal period, but also investigated the causes of metabolic dysregulation of adipose tissue in cows with ketosis. Downregulation of PCK1, a key enzyme of glyceroneogenesis, may imply a decrease in lipid synthesis in the post-partum period and might be one of the reasons for the general elevation of serum NEFA in the post-partum period. However, the simultaneous downregulation of the two key enzymes (PCK1 and PCK2) of glyceroneogenesis in ketotic cows in the post-partum period indicated a more severe lipid storage disorder. This lipid storage impairment might be related to the inflammatory response in the adipose tissue in the post-partum period. Additionally, post-partum adipose tissue in cows with ketosis might also be accompanied by dysregulated tissue fibrosis (TNC), which can considerably hinder the treatment of cows with ketosis.

## Figures and Tables

**Figure 1 animals-12-02238-f001:**
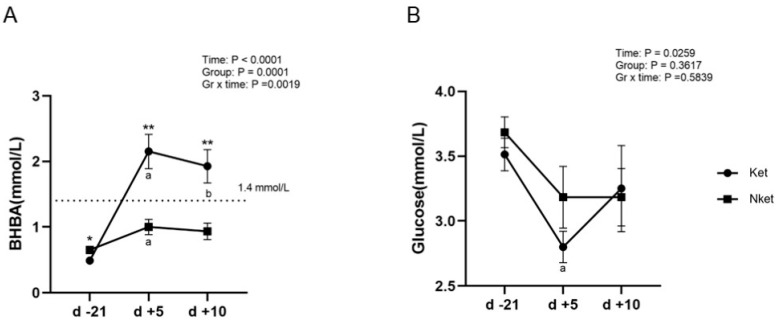
Changes in blood BHBA and glucose. Blood BHBA. (**A**) Glucose; (**B**) in ketotic cows (Ket, *n* = 8) and non-ketotic cows (Nket, *n* = 6) on d −21, d +5, and d +10 relative to parturition (LSM ± SEM). * and ** indicate *p* < 0.05 and *p* < 0.01 between the two groups at the same time point, respectively. Compared to d −21, a and b indicate *p* < 0.05 for d +5 and d +10 post-partum, respectively.

**Figure 2 animals-12-02238-f002:**
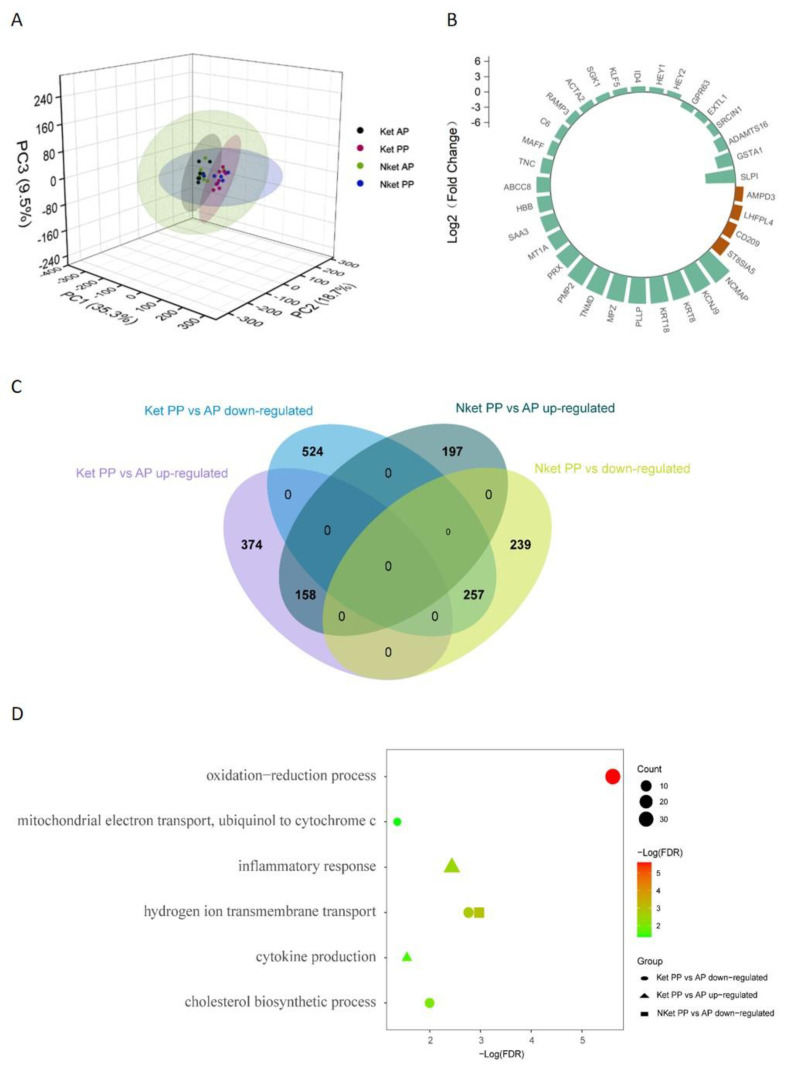
Transcriptome differential expression analysis; Ketotic cows (Ket) = 8, Non-ketotic cows (Nket) = 6. (**A**) PCA showed no separation of antepartum samples, and the effect of parturition/lactation initiation on the sWAT transcriptome was dominant; (**B**) DEGs in the Ket vs. Nket post-partum (PP) (green) and the Ket vs. Nket antepartum (AP) (brown); (**C**) Ket PP vs. AP, with 532 upregulated DEGs and 781 downregulated DEGs; Nket PP vs. AP, with 355 upregulated DEGs and 496 downregulated DEGs; (**D**) Ket & Nket PP vs. AP, GO enrichment analysis (biological process) of DEGs (FDR ≤ 0.05).

**Table 1 animals-12-02238-t001:** DEGs and fold change (FC) of sWAT transcriptome involved in β oxidation, glucose metabolism, de novo of the fatty acid synthesis, glyceroneogenesis, lipolysis, fatty acid transport, inflammation, and tissue fibrosis between d −21 antepartum (AP) and d +10 post-partum (PP) in the ketotic cows (Ket = 8, BHBA ≥ 1.4 mmol/L) and the non-ketotic cows (Nket = 6, BHBA < 1.4 mmol/L); * indicates Padj < 0.05, ** indicates Padj < 0.01.

	Ket PP vs. AP	Nket PP vs. AP	Ket vs. Nket AP	Ket vs. Nket PP
β oxidation gene network				
*CPT1A*	2.5 **	2.5 **	1.3	1.3
Glucose metabolism gene network				
*SLC2A1*	−8.5 **	−1.8	4.8	1.0
*SLC2A4*	−2.2 **	−1.7	−1.2	−1.5
*G6PD*	−3.4 **	−2.5 **	1.1	−1.2
De novo synthesis of the fatty acids gene network				
*FASN*	−38.5 **	−12.4 **	1.2	−2.6
*ACLY*	−2.8 **	−2.5 *	−1.0	−1.1
*ACACA*	−5.8 **	−3.0	1.2	−1.7
Glyceroneogenesis gene network				
*PCK1*	−3.5 *	−2.2	−1.3	−2.1
*PCK2*	−3.1 **	−1.6	1.5	−1.4
Lipolysis gene network				
*PNPLA2*	−0.8	−0.4	0.1	−0.4
*LIPE*	0.5	1.2	0.3	−0.5
*ABHD5*	0.8	0.5	0.1	0.4
*MGLL*	−0.5	−0.4	0.0	−0.1
*PLIN1*	0.3	1.2	0.1	−0.8
*PLIN2*	0.4	0.7	0.1	−0.2
Transport of fatty acids				
*LPL*	−2.7 **	−1.1	−1.1	−2.6
*CD36*	2.7 **	1.7	−1.5	1.1
*ANGPTL4*	3.2 **	4.5	1.4	1.0
*FABP4*	2.8 **	2.5	−1.1	1.0
Inflammation				
*CD163*	2.7 **	3.5 **	1.9	1.5
*CD209*	1.5	2.9 **	3.9 *	2.0
*SAA3*	−1.2	−4.0	2.5	7.9 **
*IL6*	−4.6	−4.2 *	2.1	2.0
Adipose tissue fibrosis				
*ACTA2*	−1.2	−3.5 **	−1.1	2.6 **
*TNC*	2.2	−3.6	−1.8	4.4 **

## Data Availability

Raw reads of RNA sequencing in this study were submitted to the GenBank databases under accession number: PRJNA841800.

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
