# Peer review of "Ketosis Alters Transcriptional Adaptations of Subcutaneous White Adipose Tissue in Holstein Cows during the Transition Period"

_animals, 2022, doi:10.3390/ani12172238_

Round 1
Reviewer 1 Report
The manuscript by Mao Ning et al,describes transcriptional alterations during the transition period with special reference to ketosis.
In general, the manuscript is well written and needs minor changes in the text.The critical point in my opinion of the research unfortunately is in the sample number which turns out to be quite small in the differential gene expression analyses: 14 animals divided into 2 groups.In my opinion, the number, according to the difference in BHB, should be more than 28 animals per group to have adequate statistical power.
Simple summary and Abstract: Delete in both sentences " caused by abnormal adipose tissue mobilization". Ketosis is not only generated by the mobilization and BHB is the product of oxydations. The real parameters that give you an indication for mobilization are NEFA, Tryacilglycerole or tryglicerides, and Cholesterol.
Introduction:
Lines 45-47: change "hepatic lipid accumulation" to "lipidosis." Add in the sentence that "hormonal changes" can also povocate what you describe. Example: Insulin and thyroid hormones or GnRh mediate glucose metabolism in the body - Fiore et al., 2015, Archives Animal Breeding and Fiore et al., 2015, Animal Science Papers and Reports
Lines 67-70, you mention the link between lipids and immune response, inflammatory etc. We suggest you read: Fiore et al. 2021 or Tessari et al., 2020 Animals.
In your introduction you talk a lot and rightly so about the role of NEFAs during the transition period. The question is, why didn't you also assess NEFA levels in your study? Do you have data on this?
Line 108: delete obese
Line 110: please use pre-partum or post-partum, or after calving and not afterbirth in all the text.
Line 111-112: rephrase the sentence.
Lines 115 -119: Have you analyzed BHB and Glucose Lab. as well? Unfortunately, the sensitivity and specificity of the portable machines is not very high and it is not the golden test.
Line 124:- Describe the technique you used to collect the fat. I suggest to read the paragraph tissue collection of this article (Expression of selected genes related to energy mobilisation and insulin resistance in dairy cows - 2017)
Have you received the ethical opinions for blood and fat sampling from your Institution? Is it possible to receive them?
Lines 255-261, this is very interesting, however there is transcriptomics and phenotype there is still a lot of difference. In fact, even if the gene is increased the metabolites to synthesize it may not be available. We suggest: Lisuzzo et al. 2022 https://doi.org/10.1038/s41598-022-06507-x
Please, make corrections to these minor methodological errors and test editing in all the manuscript.
Author Response
Thank you very much for your patient review and suggestions, we have modified the manuscript and the following are the replies:
Sample number: In our research, RNA-seq data analysis is the focus. For RNA-seq data analysis, two most popular methods are DESeq2 and edgeR, and we used DESeq2 to identify differentially expressed genes (DEGs). For RNA-seq data analysis, a highly cited review paper about RNA-seq analysis give the suggestion that at least 3 replicates per condition are needed (Conesa et al., 2016, Genome biology. A survey of best practices for RNA-seq data analysis). A paper published in Genome Biology(https://doi.org/10.1186/s13059-022-02648-4)also think that DESeq2 and edgeR are not suitable for large sample RNA-seq differential expression analyses (n >8).
Simple summary and Abstract: Have deleted "caused by abnormal adipose tissue mobilization" in both sentences.
Lines 45-47: Have changed "hepatic lipid accumulation" to "lipidosis." and have Added "hormonal changes" in the sentence.
Lines 67-70: Serum NEFA was also detected in the experiment, but compared with other literatures, the data results were very different and the detection was not accurate, so NEFA data was not included in the article.
Line 108: Have deleted obese.
Line 110: Have used pre-partum and post-partum in all the text.
Line 115-119: Because the trial was conducted on a dairy farm, where conditions were very limited, we needed to know BHBA and GLU results very quickly on the day of surgical trials for fat sampling, not 24 hours later. And Karapinar et al., 2020 Journal of Veterinary Internal Medicine (https://doi.org/10.1111/jvim.15794) and Fiorentin et al., 2017 Medicina Veterinária Preventiva (https://doi.org/10.1590/S1519-99402017000300004) prove that the result measured by a hand-held blood ketone meter (FreeStyle Optium Neo H-ketone Ltd.; Abbot Diabetes Care meter, Witney, Oxon, UK) is trustworthy.
Line 124: Have added the description about how we collected the fat. We have received the ethical opinions for blood and fat sampling from our institution. Here are the details in line 375-379 and China Agricultural University Laboratory Animal Welfare and Animal Experimental Ethical Inspection Form that we have submitted to the editor is in the attachment.
Line 255-261: Recommended literature has been read. The conclusion of the article is just a preliminary guess and needs to be further verified.
Thank you again for your patient review. We have resubmitted the latest manuscript and China Agricultural University Laboratory Animal Welfare and Animal Experimental Ethical Inspection Form to the editor.

Reviewer 2 Report
This manuscript presents results of an RNAseq study performed on adipose tissue from cows with ketosis and matching controls. The authors present transcriptomic analyses that identify associations between ketosis and the expression of genes PCK1, PCK2, and CD202. Results are of interest to the field, but there are issues that need to be addressed: 1) The authors must share the databases with the scientific community. Depositing the results in a repository is a must if authors want to help the field move forward. GEO is an option, but there are other data repositories. 2) The main conclusion of the study (lines 266-267) is not supported by the study's results. An RNASeq study cannot provide enough evidence to identify a single group of genes as the direct cause of ketosis. 3) the term lipodescomposition is not appropriate. Use lipolysis and lipogenesis instead.
Specific comments:
Line 14-16: With a transcriptomic analysis. it is impossible to establish if glyceroneogenesis was impacted functionally. Please rephrase accordingly.
Line 27: Again, without functional assays is not possible to establish a cause effect relationship between PCK1 activity and lipid synthesis. It is very unlikely that this response is due to a single gene.
Line 52: Define short postpartum period
Line 61. Define the threshold here (again)
Line 66: lipolysis and lipogenesis are better terms. Decomposition is not appropriate, it implies other concepts (rancidity for example)
Line 86. Although semantics, decomposing is not an appropriate term as lipolysis and lipogenesis are very specific to the processes occurring in the adipocyte. Decomposing may refer to the breakdown of fats outside the body for example, rancidity.
Line 107: Cow populations (ket nket) need more details. Include parity mode, range. BCS mode range.
Line 108: There is no accepted definition for obesity in cows. Overconditon may be the appropriate word.
Line 113: Indicate the time of blood collection relative to feeding.
Line 119: Describe the clinical condition of cows with ketosis.
Line 125. Describe in detail the biopsy process.
Line 145: Did authors deposit the data in a repository? This is highly encouraged to contribute to the field, GEO is an option, but other repositories are available.
Line 224: Are you implying that lipolysis is not responsible for NEFA release from AT? This ignores the high lipolytic rate observed in cows during the first two weeks after parturition. Other factors are ignored in your discussion including low insulin sensitivity and a state of insulin resistance in adipocytes. These two decrease lipogenesis probably independently of PCK1. This is an overinterpretation of your results, please modify.
Line 227-228: This is totally expected. Gene expression of lipolytic enzymes is not affected or even downregulated after calving. However, its activity (enzymes) is upregulated. This is why the majority of studies evaluating lipid mobilization assess phosphorylation and total abundance of HSL and ATGL. Lipolysis is controlled posttranscriptionally, therefore, the experimental design of this study does not allow the authors to draw any conclusion regarding lipolysis.
Line 246-248. The inflammatory process in ketotic cows has been characterized with immunohistochemistry and flow cytometry in the past...
https://doi.org/10.3168/jds.2015-9370
Line 266: The authors cannot make this conclusion based on a RNAseq study. Please address accordingly
Author Response
Thank you very much for your patient review and suggestions, we have modified the manuscript and the following are the replies:
- The databases have beensubmitted successfully to the GenBank databases under accession number: PRJNA841800;
- For reasons of economic and practical feasibility, we did not continue to test at the protein level, and the results were only reasonable guesses based on transcriptome analysis;
- The term lipodescompositionhas been deleted, and has used lipolysis and lipogenesis instead.
Line 14-16: Have changed the result to the transcriptome results, and added ”possibly” in the sentence.
Line 27: Have modified the meaning of the sentence into a conjecture.
Line 52: Have deteted “short”, and added “shortly”.
Line 61: Have added the threshold “BHBA≥1.4 mmol/L”.
Line 66: Has been amended to lipolysis and lipogenesis.
Line 86: The word “lipodecomposing” has been amended to lipolysis.
Line 107: Both ket group and nket group are all selected according the same details in line 143 -150. The screening conditions were the same in both groups, including parity mode range, BCS mode range.
Line 108: Have changed obese to overcondition.
Line 113: Have added the sentence: All experimental operations were carried out in the morning before feeding.Tail root venous blood was collected...
Line 119: Have added the sentence: Most of the ketosis group belong to subclinical ketosis without obvious clinical symptoms.
Line 125: Have described in detail the biopsy process in line 164-168.
Line 145: Yes. The databases have been submitted successfully to the GenBank databases under accession number: PRJNA841800.
Line 224 and Line 227-228: Have modified and added some sentences in line 305-308. such as: lipolysis is also controlled by phosphorylation, so the status of lipolysis cannot be rigorously assessed.
Line 246-248: In the article you recommeded(https://doi.org/10.3168/jds.2015-9370), the group standard is abomasum displacement, not the Type‖Ketosis. But in our research, dairy cows with displaced abomasum have been removed, so the article (https://doi.org/10.3168/jds.2015-9370) can be referenced moderately, it cannot be used as the support basis for the conclusion in the text.
Line 266: Have modified the meaning of the sentence into a conjecture, added “might be related to”.
Thank you again for your patient review. We have resubmitted the latest manuscript to you and the editor.

Round 2
Reviewer 2 Report
The authors need to upload the dataset of the RNAseq to a publicly accessible repository. The accession number that they provide in the response to reviewers is not accessible. Furthermore, the accession number and the URL of the repository need to be provided in the manuscript in the statistical methods or RNAseq sections. If the authors do not make their dataset public, this article does not benefit the audience of the journal and the scientist in the field.
Regarding the manuscript, the text has improved, however the additions need to be edited by a native English speaker that has expertise in animal agriculture. For example terms such as tail root and retained afterbirth are not use in the dairy literature. Please use accepted anatomical and clinical names such as coccygeal vein and retained fetal membranes.
The concept that reduced lipogenesis is the cause of ketosis is deeply problematic as it ignores the well proven pathogenesis factors of the disease that include lipolysis dysregulation. The authors suggest that lipolysis is regulated transcriptionally when different research groups in different countries demonstrated that lipases are controlled post-transcriptionally, unlike lipogenesis networks. The authors need to clarify this in the introduction, discussion, and conclusion.
Minor comments:
Line 24: Tailhead is more commonly used
Line 27: Again, this is a transcriptomic study. The authors can not propose PCK1 as one of the reasons for ketosis. PCK1 was found to be in a strong association with clinical ketosis.
Line 37: The authors need to discuss if the downregulation of lipid synthesis is a consequence of ketosis or the cause. It is expected that cows with ketosis mobilize a lot of fat pre-calving.
Line 74: monogastric
Line 78: and also, and probably more important, because the cow is a ruminant
Line 87: But lipolytic activity is not downregulated, in fact, is increased. Lipolysis is regulated post-transcriptionally. Please correct this concept.
Line 87-89. This is debatable, and needs to be stated that lipolysis is regulated post-transcriptionally while lipogenesis is regulated at the transcription level. Your statement is ignoring that periparturient cows are in a negative energy balance and therefore need to mobilize fat. Consequently, they will reduce lipogenesis.
Line 107: overconditioned
Author Response
Thank you again for your patient review and suggestions, we have remodified the manuscript and the following are the replies:
- The dataset of the RNAseq had summited to the GenBank databases under accession number: PRJNA841800. However this BioProject submission will be released on 2023-06-16 and here is the URL: https://dataview.ncbi.nlm.nih.gov/object/PRJNA841800?reviewer=iqedfkatp6siru4rhoosnehcli.We have also added these to materials and methods.
- Anatomical and clinical nameshave been replaced in the article, such as coccygeal vein and retained fetal membranes.
- We have made changes to the problematic statements about lipolysis and lipogenesis, see line 103-120 in the introduction, line 322-331 in discussion and line 382 in conclusion for details.
Line 24: have changed to tailhead
Line 27: The statement has been modified as you suggested.
Line 74 and line 78: This sentence does not apply well to the article and has been deleted.
Line 37, line 87 and line 87-89: For the problematic statements about lipolysis and lipogenesis, We have made the following changes in lines 103-120. The details are as follows: Studies on the subcutaneous white adipose tissue (sWAT) in perinatal dairy cows have found that the mRNA expression of genes that control adipogenesis and key enzymes in the process of adipogenesis is sharply decreased in the early lactation stage. The activity of lipogenic enzymes is controlled by transcriptional mechanisms and affected by the availability of energy [13]. Negative energy balance at the beginning of lactation leads to a pronounced decrease in the expression and activity of genes encoding proteins of the de-novo lipogenic, and glycerol-3-phosphate pathways. Reduced lipogenic activity in AT may contribute to the increase in circulating FFA levels in the immediate postpartum period. At the same time, the mRNA expression of key lipolysis enzymes was also down-regulated after delivery while lipolysis is also primarily modulated by posttranscriptional control mechanisms and the rate of phosphorylation is increased post-partum [14]. The broadly decreased transcriptional regulation of the lipogenic gene network suggests that decreased lipogenesis is an important contributor to NEFA release from sWAT postpartum. In other words, the downregulation of the lipogenesis gene network, rather than an increase in lipolysis, may be crucial for increasing the NEFA levels in post-partum blood [15-19]. However, the studies could not explain why NEFA levels in the blood are higher in post-partum ketotic cows than in non-ketotic cows. Therefore, this article will focus on the relationship between downregulation of adipogenesis and high NEFA levels post-partum.
Line 107:have changed to overconditioned.
We have attached the latest version of the manuscript to you for your review of the revised manuscript. Thanks again for your patient advice.
